behaviour/ecology/psychology

reproducibility, data decay, social learning, meta-science

**Author for correspondence:**
Riana Minocher
e-mail: riana_minocher@eva.mpg.de

# Estimating the reproducibility of social learning research published between 1955 and 2018

Riana Minocher, Silke Atmaca, Claudia Bavero, Richard McElreath and Bret Beheim

Department of Human Behaviour, Ecology and Culture, Max-Planck Institute for Evolutionary Anthropology, Leipzig, Germany

RM, 0000-0002-4614-1859; BB, 0000-0003-4653-3155

Reproducibility is integral to science, but difficult to achieve. Previous research has quantified low rates of data availability and results reproducibility across the biological and behavioural sciences. Here, we surveyed 560 empirical publications, published between 1955 and 2018 in the social learning literature, a research topic that spans animal behaviour, behavioural ecology, cultural evolution and evolutionary psychology. Data were recoverable online or through direct data requests for 30% of this sample. Data recovery declines exponentially with time since publication, halving every 6 years, and up to every 9 years for human experimental data. When data for a publication can be recovered, we estimate a high probability of subsequent data usability (87%), analytical clarity (97%) and agreement of published results with reproduced findings (96%). This corresponds to an overall rate of recovering data and reproducing results of 23%, largely driven by the unavailability or incompleteness of data. We thus outline clear measures to improve the reproducibility of research on the ecology and evolution of social behaviour.

## 1. Introduction

Scientific results often do not reproduce or replicate. This phenomenon is widely documented [1–3], and calls for reform have spurred large-scale empirical and theoretical projects on the causes and consequences of non-replication and irreproducibility across scientific fields [4–10]. Community-wide demands such as mandatory data sharing [1], preregistration of hypotheses [11], the development of infrastructure [12–15] and the innovation of new systems of research [16] are becoming increasingly common.

Of the goals of the growing open science movement, including transparency, integrity and reliability, 'replicability' has received considerable attention among the behavioural sciences, in part owing to the 'replication crisis' of psychology [17]. Replicability may be defined as re-establishing results with new data, using similar methods [18], and is crucial to advancing scientific theory, by strengthening existing scientific claims with new evidence [19]. Indeed, replicability is a necessary component of some scientific research, for example when making claims about universal human psychology [20]. Yet replication does not solely determine the quality of research and not all results may be expected to replicate [21,22].

For research in ecology and evolution, particularly for multigenerational studies of long-lived organisms such as humans, there are huge costs to data collection. Moreover, it is impossible to engineer a repeated occurrence of a temporally specific phenomenon. The focus of this type of research is often to explain particularities, rather than make generalizations. An array of factors, from demographic transition, environmental variation, historical contingency, institutional dynamics such as market integration, along with the specificities of field observation, may thus make it theoretically irrelevant and methodologically unfeasible to replicate all patterns described [23].

While replicability may not be feasible for all research, reproducibility is an achievable, minimum goal. Reproducibility may be defined as re-establishing findings using the same data and analysis procedures as in the original research [18,24]. This entails publicly available data, or data on request, along with unambiguous analysis procedures that, when repeated, reproduce results consistent with those originally reported. By ensuring the credibility and cogency of published work, streamlining workflows, increasing productivity and widening research recognition [25], reproducibility drives cumulative research [16]. Yet, data availability declines substantially with time since publication [3], and data or code-sharing policies may be insufficient, even when enforced [26,27]. Furthermore, if data are available, poor data curation [28] and unclear analysis procedures may still hinder reproducibility [1]. Despite recognition as a basic responsibility [29,30], reproducibility appears difficult to achieve.

Previous audits of reproducibility in the biological and behavioural sciences have focused on a single type of analysis [31], assessed the implementation of a journal's policy [1,27], or evaluated specific aspects of reproducibility, such as availability of data [3] or code [26]. Here, we employ a unified, general definition of reproducibility to quantitatively estimate the reproducibility of a whole research literature. We sampled the literature on the topic of social learning—a multi-disciplinary research area that spans behavioural ecology, animal behaviour, cultural evolution and evolutionary psychology. We consider a staged, conditional process by which the results of a study can be reproduced, to identify the largest barriers to reproducibility in our sample. Furthermore, being methodologically and topically diverse, our sample permits a unique evaluation of the influence of the different types of data—i.e. different research designs (observational/experimental) and different study species (human/non-human)—included in publications, on reproducibility.

# 2. Methods

## 2.1. Sample

We sampled empirical, quantitative research relevant to the topic of social learning, including both experimental and observational work, with human and non-human study subjects, without restrictions on the earliest date of publication (electronic supplementary material). We used a combination of Google Scholar and forward and backward citation tracking to identify suitable studies, searching for terms such as 'social learning' and 'cultural transmission' (electronic supplementary material). Our final sample included 560 empirical, quantitative papers published between 1955 and 2018. This comprised 446 experimental and 114 observational studies; 183 studies included only human subjects, 12 experimental studies included both human and non-human study subjects, and the remaining 365 studies included non-human animals (non-human primates, birds, reptiles, fish and small-bodied mammals). We identified 957 unique authors, each appearing on the author list of up to 49 papers in our sample, with a median number of one paper per author.

## 2.2. Protocol

### 2.2.1. Data availability

We searched for materials to aid reproduction efforts by looking for statements about the location of data in the article, and scanning the electronic supplementary material, if available. We wrote to

corresponding authors, or first authors if corresponding authors were not available, to request materials (electronic supplementary material). We sent a single reminder about five weeks after an initial data request. We categorized each study as one of 'data available online', 'data received', 'data lost/ inaccessible to author or requester within timeframe', 'no information on data (no response)', 'no information on data (no request)'. If we were able to access data online, but received no response to our request from authors, we considered data recoverable for that study.

### 2.2.2. Results reproducibility

We evaluated results reproducibility for a random subset of studies for which we recovered data, because of constraints on the number of complete reproductions we could feasibly complete. We selected 40 studies randomly, using a sampling without replacement function in R. We demonstrate that we have sufficient power at this sample size to recover parameters (electronic supplementary material).

For each paper in the subset, we identified individually citable results from the publication abstract, assuming that these results are most likely to be cited by subsequent research. We located corresponding in-text references for each result, to establish evidence for each in the form of figures, tables or estimates. For a single paper that did not contain an abstract, we identified the main results from the results section of the paper. We used the data provided, and information in the article text, supplement and any previous correspondence with the author to reproduce the evidence listed. We did not correspond with publication authors during our reproduction attempts, that is, after we received data and/or code.

If we were unable to reproduce a result, we recorded the point of failure as one of 'data unclear— incomplete or incoherent', 'analysis unclear' or 'results disagree'. We considered data unclear if we could not use the available data to evaluate a particular result. This meant that data were missing for a particular experiment, or data were insufficiently documented, being too cryptic or too raw, to correspond to the results presented. If the data were usable, but the analysis was unclear, being under-described in the text, or too complicated or novel to implement without analytical code or substantial support, the result failed at third stage, 'analysis unclear'.

If data were usable and an analysis repeatable for a result, we evaluated consistency between published and reproduced results. If a reproduced result reversed the direction of, or negated, the reported effect, we coded the cause of failure to be 'results disagree'. For empirical frequencies or proportions, we confirmed results when they were quantitatively equivalent. For modelled results, we considered the level of detail in the original article; if the article reported only the significance of an effect, we looked for an effect with a confidence interval that did not overlap zero. If the article reported the direction and magnitude of an effect, we confirmed that the direction and magnitude of our reconstructed result were consistent with this, if not numerically equivalent, to allow for sampling variation in estimation algorithms. We reproduced all analyses in R, even if originally conducted in a different software, such as SPSS. An example of our reproduction protocol is available at https://github.com/rianaminocher/reproducibility-example.

## 2.3. Data analyses

We characterized the staged, conditional process by which results of a paper can be successfully reproduced. Reproducibility depends sequentially on (1) data recoverability, the availability of data to attempt reproduction of analyses; (2) data usability, the completeness and clarity of data, when available; (3) analytical clarity, the adequacy of published reports for repeating analyses; and (4) results consistency, the agreement of reproduced results with published results. Each stage is conditional upon the previous. Specifically, if data cannot be recovered, data usability and analytical clarity cannot be assessed, and if the data are unusable and the analysis unclear, the reproduction cannot be attempted. When a result passes the first three stages, it may still fail to reproduce because it was originally misreported.

Let $x_{s,i}$ be the outcome of stage $s \in \{1, 2, 3, 4\}$ for publication $i$. In the first stage, $s = 1$, each publication $i$ has $x_{1,i} = 1$ if data were recovered, and $x_{1,i} = 0$ otherwise.

In statistical notation:

$$x_{1,i} \sim \text{Bernoulli}(p_{1,i})$$

and

$$p_{1,i} = \alpha_{Y[i]} \exp(-\lambda_{Y[i]} v_i),$$

where $v_i$ is the number of years since publication, $\alpha$ is the probability of recovery at time of publication and $\lambda$ is the rate of decay. We estimate individual parameters $\alpha$ and $\lambda$ for each type of study $Y_i$, which is a combination of the species studied and study design (human experimental, human observational, non-human experimental and non-human observational).

We assume the probability of data recovery decays exponentially with time since publication. When any one of a large number of errors can cause a failure to recover data at any point in time, such as loss of files, retirement of archives, and technological change, the resulting decline will tend to be multiplicative [32]. Moreover, previous research shows a steep and seemingly exponential decline in data availability [3]. We conduct a robustness check on this assumption by fitting an alternative model, using a logistic link function (electronic supplementary material).

We consider the effects of study type, because we expect that data including human and non-human subjects are likely constrained by different ethical regulations. Concomitantly, experimental data may be easier than observational data to share, having been generated in a specific format and unlikely to be re-used or re-analysed by the experimenter, in contrast to observational data, which is often used repeatedly, to investigate different research questions.

When $x_{1,i} \neq 0$, the second stage, data usability, is assessed; when $x_{2,i} \neq 0$, the third stage, analytical reproducibility, is assessed; and when $x_{3,i} \neq 0$, the final stage, results consistency, is assessed. The statistical models for these stages are identical, but with independent parameters. For $s = 2$, $s = 3$ and $s = 4$, the rate of reproducibility is a function of the paper's publication date $v_i$, as in the first stage. In statistical notation, for $s = 2$, $s = 3$ or $s = 4$:

$$x_{s,i} \sim \text{Binomial}(n_i, p_{s,i})$$

and

$$\text{logit}(p_{s,i}) = \phi_s + \beta_{s,i} + \gamma_{s,v[i]}.$$

The parameters $\beta_{s,i}$ represent the paper-specific intercepts, drawn from a normal distribution with the same variance $\psi_s$. This model accounts for the correlation between results of a single paper; the reasons for failure for one result of a paper may cause failure of another result (such as poor data documentation). Conversely, the benefits to reproducibility of a result through adoption of specific research practices (such as clear code comments) may affect the reproducibility of other results in the paper.

The parameters $\gamma_{s,v[i]}$ represent the age-specific intercepts. We do not assume any particular functional form for the relationship with $v_i$, instead using a Gaussian process smoother (electronic supplementary material). This allows us to increase the power with which we estimate the influence of age, and is justified as we did not have a theoretically or empirically motivated expectation that results reproducibility will decay in the same way as data availability.

We combined probabilities of reproducibility at each stage, conditional on previous stages, to estimate an overall success rate for a publication (electronic supplementary material). That is,

$$p(r) = (p_1)(p_2)(p_3)(p_4),$$

where $r$ is the event that any particular result reproduces.

We conducted analyses in R (v. 4.0.3 [33]). We fit our model using the Stan engine, implemented in R with rstan (2.21.2 [34]). We use regularizing priors to constrain parameters to plausible values, and validated these using prior predictive simulation (electronic supplementary material). Results presented are summaries of 10 000 iterations of 4 chains. We assessed convergence and mixing with the 4th version of the $\widehat{R}$ convergence diagnostic as implemented in Stan [35], the estimate of the autocorrelation-adjusted number of samples (n_eff), and visual inspection of the trace plots of all chains (electronic supplementary material). All intervals reported reflect the narrowest width of the posterior that contains 89% of the probability mass [36]. We provide data and analysis code to reproduce our reported results at https://github.com/rianaminocher/reproducibility-analysis.

# 3. Results

## 3.1. Data recovery

Through direct correspondence and searching for materials online, we recovered data for 167 studies, 30% of our full sample of 560 papers (figure 1). Data were available online—in electronic

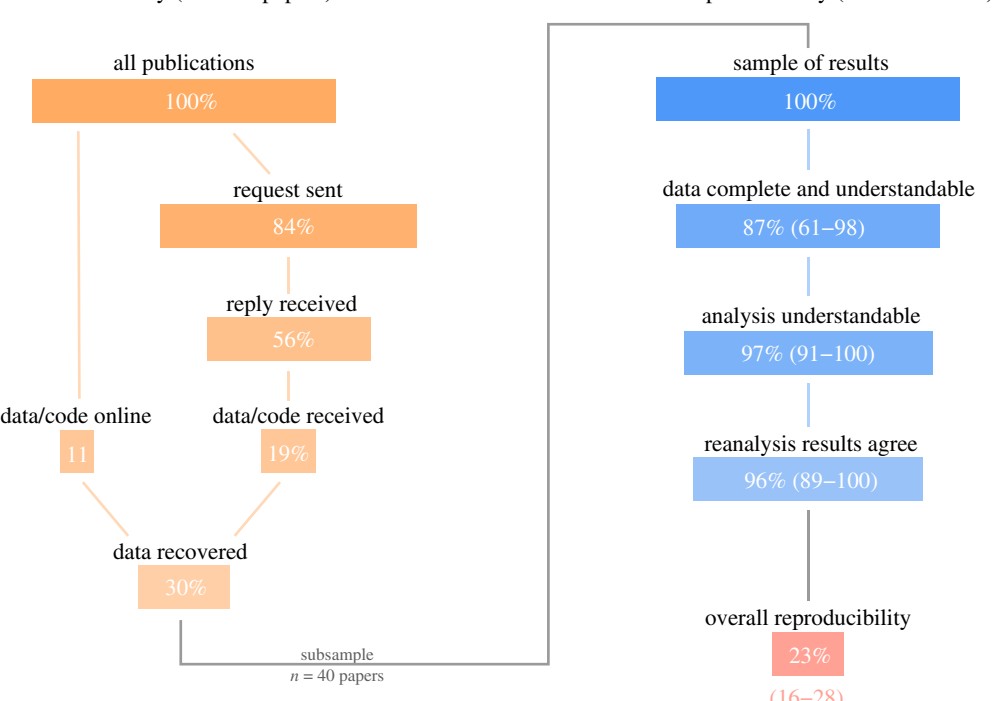

data recovery (*n* = 560 papers)    results reproducibility (*n* = 111 results)

**Figure 1.** The left side of the figure shows the process of 'data recovery', while the right side of the figure shows the process of 'results reproducibility'. The width of each bar represents the proportion of the sample at each stage of the process, i.e. the number of papers, in the stages of 'data recovery', and the number of results, in the stages of 'results reproducibility'. (*a*) Of 560 publications surveyed (100% of the sample), we recovered data online for 62 (11%). Of the remaining papers, we sent requests to authors of 473 papers (84%), and received a reply in 315 cases (56%). Through correspondence, we received data for an additional 105 publications (19%). Thus, we categorized a total of 167 studies with data available, 30% of our initial sample. We sampled 40 of these 167 publications randomly, to evaluate subsequent stages of reproducibility. (*b*) From 111 results, we identified in these 40 publications (100% of this sample of results), we estimated the probability of data usability, given data are available, to be 87% (83/111 results). The probability of analytical clarity, given data are available and usable, is estimated to be 97% (77/83 results). The probability that reproduced results correspond to reported results, given data are available and usable, and analysis clear, is estimated to be 96% (73/77 results). In (*b*), the numbers in parentheses correspond to the intervals of the estimated conditional probabilities ($p_2$, $p_3$, $p_4$). This results in an overall probability of reproducibility ($p_r$) across all stages of 23%.

supplementary material, data repositories or in the main article—for 62 of these papers (11%). Of the remaining 492 publications, we had sent e-mails requesting data, to the authors of 473 publications. We received a response in 315 cases (56% of the total). We received data from authors for 105 of these publications. This corresponds to a total of 167 publications with available data. For 39 studies, authors indicated willingness to participate in our study, but we did not receive materials. In nine cases, authors indicated access to relevant files, but unwillingness to participate in our study.

## 3.2. Reproducibility of results

Of 167 studies that we were able to recover data for, we randomly selected 40 to estimate subsequent stages of reproducibility. Within these 40 studies, we identified a total of 111 results, with a median of 3 and a maximum of 6 results per study.

Of 111 results, we determined the data to be usable and clear in 83 cases. Cases that failed included data files in incomprehensible formats, such as Microsoft Excel tables with multiply labelled or duplicated sheets. In other cases, data were missing, upon examination of materials. That is, the data for just one of two experiments of a publication were available to us, or the data provided were aggregated for presentation and unusable for the purposes of reconstructing a result.

Of 83 results which corresponded to usable data, we determined the analysis to be clear and repeatable in 77 cases. We recorded analyses to be unclear when no code was available, but the publication referenced bespoke scripted techniques, which we determined to require numerous

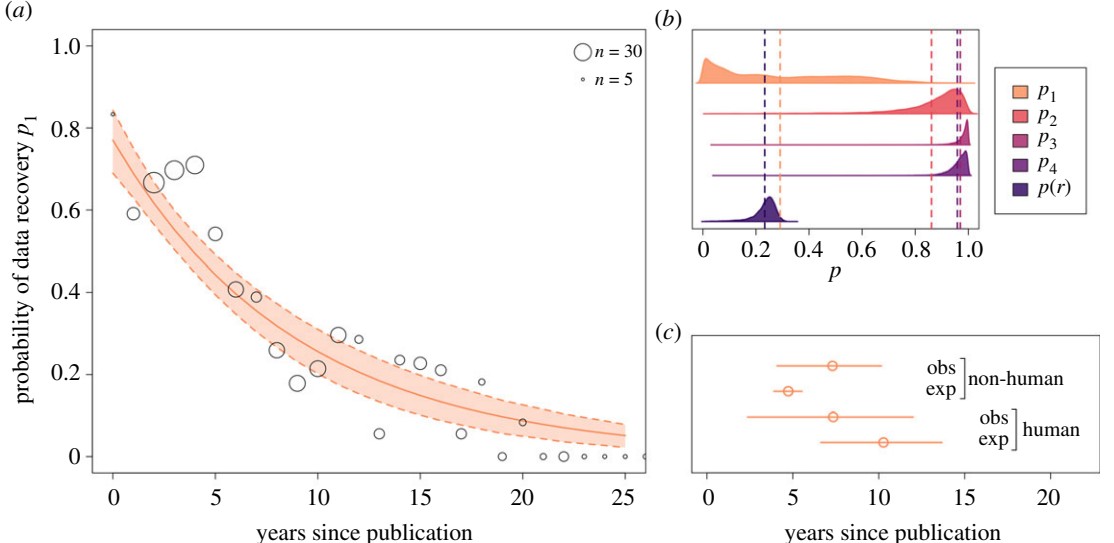

**Figure 2.** (*a*) The predicted probability of data recovery declines exponentially with increasing time since publication, halving every 6 years. The solid line plots the expected exponential decay curve and the shaded interval between dotted lines shows the 89% compatibility interval. The empty circles plot the observed data, i.e. raw proportion of studies, for each year, for which we obtained materials. The size of each circle is scaled by the total number of observations for that year. The *x*-axis is truncated at 26 years since publication, for clarity of presentation, as the expected probabilities beyond this point are <0.01. (*b*) Posterior prediction densities, marginalized across ages of papers in the sample, for $p_1$, the probability of data recovery; $p_2$, the probability of data usability, conditional on recovery; $p_3$, the probability of analytical clarity, conditional on data usability and recovery; $p_4$, the probability of reproduction, conditional on analytical clarity, data recovery and usability; and $p(r)$, the combined probability of reproducibility across all four stages of data recovery, usability, analytical clarity and final reproduction. The dotted lines plot the mean of each distribution. (*c*) The data half-life, decomposed by study type, indicates that human experimental data decay most slowly. The point plots the mean expected half-life value for each data type, and the line indicates the 89% compatibility interval.

algorithmic assumptions that we did not have sufficient information to make. Similarly, we recorded analyses to be unclear when highly unprocessed data (raw video files) were available to us, with code files in multiple languages to process these, but without relevant directions on how to apply these to the raw data. We also recorded analyses as unclear when we could not fit models described by authors, even with available (non-executable) code.

When repeating the analyses for 77 results with clear analyses and data, we found the reconstructed result to correspond to the published findings in 73 cases. For three of four results which did not reproduce, our reproduced model estimates reversed or negated the effect stated in the published result. For one result which did not reproduce, we were unable to reproduce the magnitude and significance of the published correlation coefficients, using a set of non-parametric statistical tests described.

## 3.3. Overall rate of reproducibility

We estimated $p_1$, the probability that data from a paper in our sample is recovered, to be 0.29 (CI: 0.27–0.32). We estimated $p_2$, the conditional probability that data from a paper are usable, given that they are recovered, to be 0.87 (CI: 0.61–0.98). We estimated $p_3$, the conditional probability that analyses are understandable, given that data are available and usable, to be 0.97 (CI: 0.91–1.00). We estimated $p_4$, the conditional probability that reproduced results agree with published results, given that data are available, usable and analyses are understandable, to be 0.96 (CI: 0.89–1.00). Thus, we estimate an overall success rate for a publication of 0.23 (CI: 0.16–0.28). We find that success rates by papers tend to be highly bimodal, with either most results or few results of a specific paper reproducing. Accordingly, the variance between individual papers is relatively high (electronic supplementary material).

## 3.4. Data decay

The availability of social learning data decays exponentially with time since publication (figure 2*a*). The expected probability of finding material for any publication halves every 5.7 years on average

(CI: 4.2–7.1). Consequently, the probability of recovering data for studies published more than 20 years ago is close to zero (figure 2a; expected probability less than 0.01 at $t = 23$ years). Human experimental data decays most slowly, compared with other data types (figure 2c). The estimated half-life for human experimental data, of 9.6 years (6.1–12.8), is substantially larger than half-lives ranging from 5 to 6 years for other data types (human observational: 6.1 (2.0–9.2); non-human observational: 4.5 (3.8–5.3); non-human experimental: 6.5 (4.0–9.0)).

While age steeply influences data recovery, we found little to no effect of age on data usability, analytic clarity and reproduction of results. This was not unexpected—because data recovery is strongly influenced by publication age, it follows that the sample of papers with available data are already conditioned by age (figure 2a). When decomposing the overall publication success rate by age of publication, we estimate the overall probability of success in the year 2018 to be 0.68 (CI: 0.50–0.86), substantially higher than when marginalized across dates in the sample at 0.23 (CI: 0.15–0.28), due to the higher probability of data recovery for recent publications.

# 4. Discussion

Under one in four reproduction attempts in the social learning literature succeed, currently. Our estimates are contingent upon the age distribution in our sample, our reproducibility protocols and our four-stage definition of reproducibility. Given this, we expect that reasonable effort to recover data will fail for about 70 of 100 potential papers. Conditional on recovering data, we expect a further 7 papers to fail to reproduce, because of unclear data, ambiguous analysis methods or reanalysis disagreements.

Data unavailability is by far the largest barrier to reproducibility in our sample. As indicated by similar research [3,26], this barrier is smallest for recent publications—the probability of data recovery in 2018 is as much as eight times higher than expected for the year 2000. Of course, sharing data in the behavioural sciences was neither normative nor feasible until recently. Before the proliferation of open-source software tools [13], data were often stored in outdated or proprietary formats. Frequently, replies to our requests included the phrase, 'unfortunately, that study was X computers ago'.

To extend data lifetime, we require institutional infrastructures, because individual researchers retire, and data accumulates over a career. Research institutions and cross-institutional collaborations now invest in data management roles and develop bespoke data processing pipelines, and academic journals regularly adopt data sharing policies [37–39]. While the benefits of increasing investment in data infrastructure relative to producing new research requires further quantitative investigation, the current measures to improve data archiving face major challenges. These include proper enforcement [26,28,40–42], including appraising files before publication [1], empowering reviewers [43] and expanded hosting for analysis code as well as data files [44].

Most researchers publishing in ecology and evolution are fluent in computational tools, but not necessarily in data science skills [45]. The use of plaintext data formats ensures stable, long-term accessibility of data, and free hosting services are available through the Open Science Framework, GitHub and Data Dryad. Numerous resources are available, being developed, and increasingly used to improve data science fluency, such as Software Carpentry (http://software-carpentry.org), R for Data Science [46], The Practice of Reproducible Science [47] and FAIR (Findable, Accessible, Interoperable and Reusable) [13], among others [48–50].

Maintaining *data provenance* is central to reproducibility. In principle, if data are to be re-used, the entire history of the data and their origins, from data collection to transcription, coding, transformation and analysis, should be documented and archived. This is facilitated by version control systems (e.g. git). In practice, this is difficult to achieve, because of both ethical and technical constraints on sharing raw data. Re-processing raw video or audio files, while theoretically possible, can be prohibitively difficult, demanding an unreasonable amount of effort. Unless open-source script is available to transform raw data into processed for analysis, these data are unusable for the purposes of reproducing the results of a publication.

Yet, highly processed data can fail to correspond to analyses presented in an article, for example when aggregated into categories, rather than a single entry per observation [51]. The meaning of variables can become impossible to decipher over time, given frequent updates to variable descriptions for presentation purposes, or creation of new, very similar variables. This is especially true for longitudinal data, where new data are continuously added, thus easily conflated with older data. That individual-level data be archived, in a tabular format, along with a data dictionary or variable key in the language of publication (for our study, English) is not trivial practice.

Authors of papers using longitudinal, observational datasets expressed fears that published data may be wrongly re-interpreted and thus bolster false or contradictory analyses (but see [52]). Data can be extremely costly to generate, under many circumstances. Still, numerous solutions to issues of data privacy and rightful credit exist, including data publishing embargoes, password-protected databases and citable DOIs [53]. A related concern highlighted in our study was that data are extremely costly to prepare for sharing upon request. Yet ostensibly, materials should be ready to share at the review stage of the publication process, as reviewers regularly request access to the data. Widespread data science tools and data archiving technology dramatically reduce the costs of data curation today [54]. Indeed, if data are well archived, well documented and made available at the time of publication, it should demand negligible time and effort to deal with, or obviate the need for, a data request.

That data for recent publications are available, yet reproducibility fails due to a lack of analysis clarity, suggests that shifting research attention towards the impact of code availability and analysis reporting will be extremely valuable [48]. Recent analyses are often documented in statistical scripting languages, frequently R [26], as well as Stata, Python, Mathematica and Matlab, but authors are hesitant to make code public. Even when non-functional, code (in any programming language) facilitated our ability to reconstruct results. Messy code is better than no code [55], as it can document data manipulation or exclusion that is otherwise opaque, clarify the sequence of analyses conducted, and record any algorithm assumptions. These details are likely too complex to compress, even into a detailed supplement. Moreover, an early commitment to share code can improve the way that code is written, encouraging the use of 'linting' for code readability [56] and explanatory code comments, facilitating reproducibility.

A relatively high rate of response to a survey, even when materials were not recoverable, and general support for the goals of our study, is consistent with previous work that indicates researchers readily accept reproducible norms and reward reproducible practices [57–59]. We interpret our finding that human experimental data have a longer lifetime than other data types to be indicative of earlier adoption of reproducible research norms in the field of psychology; the 'Replication Crisis' has also been called the 'Credibility Revolution' [60]. The growing recognition of the importance of reproducibility is encouraging for researchers, particularly junior scholars, who often consider making requests for materials from published work.

The decay rate of social learning data, while alarming, is not entirely unique. Our estimate of approximately 30% data recoverability resembles findings of 38% [61], 32% [62], 26% [63,64] and 19% [3]. Some estimates for more recent publications find data recoverability to be slightly higher at 56% (articles published in the year 2015) [65] or 58% (articles published 2014–2018) [66]. Our estimate of results reproducibility, conditional on previous stages, however, is more difficult to quantitatively compare against similar audits, which have found reproducibility of results to be anywhere between 83% [67], 70% [31,65], 68% [64], 58% [66] and 1.1% [68] of surveyed publications and/or results.

In relation to this, we consider that the extent to which an estimate of reproducibility is relevant should depend on how reasonable the reproduction efforts undertaken were. A major challenge to establishing a useful estimate of reproducibility is that some researchers may succeed to reproduce a result where other researchers fail. This can be because of a lack of access to necessary software (proprietary programs) or hardware (server clusters), or a lack of necessary skill or knowledge for a specialized analysis. Given this, there may be no universal standard with which to label a result 'non-reproducible'.

Because the skillsets of a reasonable researcher can only be defined relative to the norms of a particular scholarly community, we suggest that any reproducibility effort has to be defined in similar relative terms. Throughout our study, we attempted to remain conscious of the time, effort and skillset that a typical early-career researcher in the evolutionary behavioural sciences might invest in a reanalysis (the reasonable researcher criterion). We sampled papers published in English, used institutional access to software and server clusters, researched methods we were unfamiliar with, and endeavoured to correspond with authors of publications in a formal, professional tone. This is central to our definition of a successful reproduction.

Our model of reproducibility aims to capture the process underlying a successful reproduction. Data need to be available and usable, analyses repeatable and results consistent, to consider published results reproducible. Still, we acknowledge that our model and design prohibits an analysis of the sets of papers which failed at any stage. Consider the counter-factual reality—we identify a result that failed to reproduce, because the data were lost. Then we go back in time and intervene, such that the data are saved. The expected probability that this result is reproducible may not be the same as a result for which the data were recovered, in reality, because there may be associations between data

recoverability and subsequent stages of reproducibility. We have only measured subsequent stages of reproducibility for results for which data exist, that is, conditionally, so we cannot estimate the causal influence of results reproducibility unconditionally.

Finally, our definition of reproducibility focused on reproducing the results of a publication, and was thus best facilitated by data-sheets processed for analyses. We thus acknowledge that our protocol cannot evaluate upstream errors or analysis procedures, prior to the data provided to us. Most importantly, our assessment of reproducibility does not appraise the overall quality of research. In reconstructing analyses exactly as presented in the paper, we did not critique appropriateness of methodological or analytical choices for particular results, even when we ourselves might have collected or processed the data differently. Such a reproduction procedure is complementary to, but no substitute for, criticism of the scientific value of a published result or approach.

Reproducibility of empirical work is a minimum requirement for research—a necessary precursor to replication, meta-analysis or further theory development. Much of the data in social learning research originates in populations that are undergoing rapid social and environmental change—the significance of these data can only appreciate with time. Our findings underscore the importance of motivating and maintaining efforts to secure data for posterity. Further to this, our research implies that the rapid adoption of reproducible norms and tools of data sharing should slow data decay tremendously, such that a study performed in a decade will show a very different decay rate, and correspondingly a much higher reproducibility rate, than estimated here.

Acknowledgements. We thank Anne Büchner, Leonie Ette and Kristina Kunze for collecting data, Dr Cody T. Ross for consulting on analyses and model parameterization, and the Department of Human Behaviour, Ecology and Culture at the Max-Planck Institute for Evolutionary Anthropology for helpful discussion and feedback on the project at various stages. We thank Dr Alfredo Sánchez-Tójar and Dr Antica Culina for feedback on an earlier version of the manuscript. We thank the two anonymous reviewers who reviewed our work during this difficult time. Finally, we would like to thank all the authors of the work we sampled, who took the time to correspond with us, and to dig up, clean, or comment old data/code for us.

Data accessibility. Data and relevant code for this research work are stored in GitHub: https://github.com/rianaminocher/reproducibility-analysis and have been archived within the Zenodo repository: https://doi.org/10.5281/zenodo.5226241 [69].

Authors' contributions. R.M. designed the study, collected data, conducted analyses and drafted the manuscript. S.A. coordinated data collection and provided feedback on the manuscript. C.B. collected data, corresponded with authors and provided feedback on the manuscript. R.M. provided feedback on statistical analyses and detailed feedback on the manuscript. B.B. designed the study, participated in analyses and provided detailed feedback on the manuscript.

Competing interests. We declare we have no competing interests.

Funding. This research was funded by the Max-Planck Institute for Evolutionary Anthropology.

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
