## [Peer Review File · Royal Society Open Science]

Review History

RSOS-210450.R0 (Original submission)

Review form: Reviewer 1

Is the manuscript scientifically sound in its present form?

Yes

Are the interpretations and conclusions justified by the results?

Yes

Is the language acceptable?

Yes

Do you have any ethical concerns with this paper?

No

Have you any concerns about statistical analyses in this paper?

No

Recommendation?

Accept with minor revision (please list in comments)

Comments to the Author(s)

****Disclaimer:** The authors contacted me for data for some of my past social learning research, so my responses and studies would be included in their dataset / analysis. However this would apply to everyone in the field of social learning, and I had no input into the study in any other respect, so I don't consider this a conflict of interest.**

I really like this study of the reproducibility of findings in social learning research. It contributes both to the ongoing efforts to improve the scientific rigour of the behavioural and evolutionary sciences, and to the study of social learning in humans and non-human species. The methods are robust and well-documented, the results are adequately analysed and reported, and the conclusions are constructive and coherent. It's valuable and novel to know that the biggest hurdle to reproducibility is lack of availability of data, and encouraging that if data are present then a large majority of results are reproducible (and also that data availability is improving all the time). I recommend publication, with some minor suggestions / requests for clarification (in order of appearance in the ms, not importance):

Page 2, line 45 "relevant to the field of evolutionary anthropology": this kind of goes against the subsequent sentence which highlights the multidisciplinary relevance of social learning to other disciplines like behavioural ecology, evolutionary psychology etc. Suggest deleting or moving evolutionary anthropology to the multidisciplinary list of fields.

Page 3, line 11: this is in the SI but I would add here for clarity that synonymous or similar terms like "cultural transmission" were also used as search terms

page 6, line 11: an aside really - I understand why the papers that were selected for analysis are unidentifiable. However, I wonder if the authors of the present paper have contacted the authors of the 4 findings that failed to match the original claim? The original authors may be willing to collaborate on a re-analysis, or correct or retract the original paper. This strikes me as an important correction to the literature, if the current authors are aware that there are (potentially) false claims in the literature that are presumably still being cited, but haven't disclosed what they are.

Page 6, line 36: regarding the differences between human, non-human, observational and experimental categories: I wonder how robust these are to author differences? E.g. if author X runs human experiments, authored several papers in this sample and included data in their papers / responded to requests, while author Y runs non-human observational studies, authored several of these in the sample, and did not include data / did not respond, then these category differences could be more to do with those individual author's practices than anything inherent about the different methods and study species. This may be particularly the case for a relatively small field with some authors publishing multiple studies within it. I know there are paper-specific intercepts in the model, but I don't think there are author-specific intercepts. Can a bit more information be provided about how many papers were in each of these categories, and if the same author appeared across more than one paper, to get some insight into this?

Page 7, figure 1 caption: Why were 89% CIs used rather than 95%? ... Just kidding, I know why. But seriously, it is only mentioned in the figure caption that 89% was used, while there are several CIs in the text. If all of the CIs in the text were also 89% this needs to be mentioned somewhere.

SI: "If a study included both observational and experimental designs, we recorded it as "observational", and if a study included both human and non-human study species, we corded it as "human"." I don't really see the logic of this, it needs some justification. If a study compares

social learning in human children and chimpanzees on the same task (e.g. Dean et al. 2012), how come this is coded as 'human' and not 'non-human'? It strikes me that the chimpanzee part would present more methodological challenges than the human part, and be the bigger limiter on data analysis and recording. Perhaps a third 'human and non-human' category is needed here.

Acknowledgements: it might be nice to acknowledge all the authors who took the time to respond to email requests and dug up old data and scripts.

Review form: Reviewer 2 (Nick Tierney)

Is the manuscript scientifically sound in its present form?

Yes

Are the interpretations and conclusions justified by the results?

Yes

Is the language acceptable?

Yes

Do you have any ethical concerns with this paper?

No

Have you any concerns about statistical analyses in this paper?

No

Recommendation?

Accept with minor revision (please list in comments)

Comments to the Author(s)

I have some minor comments in my review:

Review of paper: "Estimating the reproducibility of social learning research published between 1955 and 2018"

Summary

As the title states, this paper assessed and estimated the reproducibility of a set of papers published in social learning from 1955 to 2018.

This is an excellent paper. I enjoyed reading it immensely. The methodology is sound, and clearly a large amount of effort has been put in all aspects, from the methodology itself being very time consuming and writing and responding to many emails, to the sophisticated, but not too complex statistical model, through to the discussion that brings home many key points about the current issues in reproducibility.

I also greatly appreciated that a github repository (<https://github.com/rianaminocher/reproducibility-analysis>) was provided with the code, and in particular that time was taken to write a README to explain how to run the analysis. My only thoughts further to this are that a system like targets (<https://github.com/ropensci/targets>), and renv (<https://github.com/rstudio/renv>), could be used in the authors' future work, to help make reproducibility even more stable, as well as help improve installation. I don't want the authors to

turn around and use targets and renv with this paper, but I just thought they might be interested in knowing about it.

I have only very minor comments.

Comments on the paper

Section 1, line 15

It might be useful to very briefly distinguish between the terms reproduction and replication earlier in the paper - since "replicability" is quoted in the second paragraph, perhaps that could be a good place to briefly define the two terms? I do note that replicability is defined in the third paragraph on page one, and that these terms have been defined many times in other papers, so I will understand if the authors would prefer to not do this.

Section 1, line 51

What are the different types of data? Can you provide some examples of these?

Section 2A, line 11

What does forward and backward citation tracking mean?

Section 2B, part ii, line 34

I assume that this is sample without replacement? I also assume a seed was set, but I could not find this in the GitHub repository. I guess as long as the data that was subsetted is stored from the first time this isn't a problem, so perhaps there is no need to address this in the text, but just thought it might be worthwhile to mention.

Section 2C, line 31

"We allow both alpha and lambda to vary with the species studied..."

Is "species" here a typo?

Figure 1

I found Figure 1 quite tricky to interpret. I could not understand which text belonged to which bar - for example on the left hand side, does the top bar belong to "request sent 84%"? If so, then what text belongs to the bottom bar? Is it "Data/code received 19%"? or "Data recovered 30%"?

Another reason this figure is hard to understand is that the bars are unaligned to an axis, so it is hard to compare and understand them (see <http://visiphilia.org/2016/08/03/CM-hierarchy> for more details on Cleveland-McGill's graphical hierarchy).

Since I believe the main goal of this visualisation is to communicate the flow of the number of surveyed, and where those numbers decreased from 100%, I think a sankey or alluvial chart might be useful? I would recommend potentially rebuilding it with ggalluvial (<https://cran.r-project.org/web/packages/ggalluvial/vignettes/ggalluvial.html> - an example - <https://mdneutzerling.com/post/my-data-science-job-hunt/>).

If an alluvial chart is not desirable, it would at least be helpful in the figure text to describe which way (top to bottom, bottom to top) to read the figure.

Figure 2A

Figure 2A should contain a legend to indicate what the bubble size represents.

Decision letter (RSOS-210450.R0)

Dear Ms Minocher

On behalf of the Editors, we are pleased to inform you that your Manuscript RSOS-210450 "Estimating the reproducibility of social learning research published between 1955 and 2018" has been accepted for publication in Royal Society Open Science subject to minor revision in accordance with the referees' reports. Please find the referees' comments along with any feedback from the Editors below my signature.

Additionally, please note we require all authors to have an active email address that is able to receive messages from the journal; at present, silka_atmaca@eva.mpg.de is not receiving messages - please can you confirm an alternative email address with your colleague or ask them to 'white list' messages from the journal?

Please submit your revised manuscript and required files (see below) no later than 7 days from today's (ie 13-Aug-2021) date. Note: the ScholarOne system will 'lock' if submission of the revision is attempted 7 or more days after the deadline. If you do not think you will be able to meet this deadline please contact the editorial office immediately.

on behalf of Prof Kevin Padian (Subject Editor)
 openscience@royalsociety.org

Associate Editor Comments to Author:

Thank you for your patience while the journal sought peer reviewers - regrettably, the last few months have been exceptionally difficult for editors seeking reviewers, so we are very grateful not only to the authors for your patience but the referees for your support at a tricky time.

The general view of the work is positive, and publication following revision is recommended.

One note for the authors from one of the reviewers is that there may be a minor bug in your code (see below), though the reviewer noted it may be an issue with their software, rather than the code per se (in any case, something to take a quick look at):

After following installation instructions, I could not replicate the paper and got this error:

...

```
Error in Module(module, mustStart = TRUE) :
function 'Rcpp_precious_remove' not provided by package 'Rcpp'
```

...

However after restarting R this error went away, and then I got this error when running the fitting code:

...

```
fit <- stan(file = "model.stan",
  data = data,
  chains = 4,
  cores = 4,
  iter = 1e4)
```

...

...

...

```
Chain 4: Iteration: 1 / 10000 [ 0%] (Warmup)
Chain 4: Iteration: 4000 / 10000 [ 40%] (Warmup)
Error in unserialize(socklist[[n]]) : error reading from connection
```

...

Reviewer comments to Author:

Reviewer: 1

Comments to the Author(s)

****Disclaimer:** The authors contacted me for data for some of my past social learning research, so my responses and studies would be included in their dataset / analysis. However this would apply to everyone in the field of social learning, and I had no input into the study in any other respect, so I don't consider this a conflict of interest.**

I really like this study of the reproducibility of findings in social learning research. It contributes both to the ongoing efforts to improve the scientific rigour of the behavioural and evolutionary sciences, and to the study of social learning in humans and non-human species. The methods are robust and well-documented, the results are adequately analysed and reported, and the

conclusions are constructive and coherent. It's valuable and novel to know that the biggest hurdle to reproducibility is lack of availability of data, and encouraging that if data are present then a large majority of results are reproducible (and also that data availability is improving all the time). I recommend publication, with some minor suggestions / requests for clarification (in order of appearance in the ms, not importance):

Page 2, line 45 "relevant to the field of evolutionary anthropology": this kind of goes against the subsequent sentence which highlights the multidisciplinary relevance of social learning to other disciplines like behavioural ecology, evolutionary psychology etc. Suggest deleting or moving evolutionary anthropology to the multidisciplinary list of fields.

Page 3, line 11: this is in the SI but I would add here for clarity that synonymous or similar terms like "cultural transmission" were also used as search terms

page 6, line 11: an aside really - I understand why the papers that were selected for analysis are unidentifiable. However, I wonder if the authors of the present paper have contacted the authors of the 4 findings that failed to match the original claim? The original authors may be willing to collaborate on a re-analysis, or correct or retract the original paper. This strikes me as an important correction to the literature, if the current authors are aware that there are (potentially) false claims in the literature that are presumably still being cited, but haven't disclosed what they are.

Page 6, line 36: regarding the differences between human, non-human, observational and experimental categories: I wonder how robust these are to author differences? E.g. if author X runs human experiments, authored several papers in this sample and included data in their papers / responded to requests, while author Y runs non-human observational studies, authored several of these in the sample, and did not include data / did not respond, then these category differences could be more to do with those individual author's practices than anything inherent about the different methods and study species. This may be particularly the case for a relatively small field with some authors publishing multiple studies within it. I know there are paper-specific intercepts in the model, but I don't think there are author-specific intercepts. Can a bit more information be provided about how many papers were in each of these categories, and if the same author appeared across more than one paper, to get some insight into this?

Page 7, figure 1 caption: Why were 89% CIs used rather than 95%? ... Just kidding, I know why. But seriously, it is only mentioned in the figure caption that 89% was used, while there are several CIs in the text. If all of the CIs in the text were also 89% this needs to be mentioned somewhere.

SI: "If a study included both observational and experimental designs, we recorded it as "observational", and if a study included both human and non-human study species, we corded it as "human"." I don't really see the logic of this, it needs some justification. If a study compares social learning in human children and chimpanzees on the same task (e.g. Dean et al. 2012), how come this is coded as 'human' and not 'non-human'? It strikes me that the chimpanzee part would present more methodological challenges than the human part, and be the bigger limiter on data analysis and recording. Perhaps a third 'human and non-human' category is needed here.

Acknowledgements: it might be nice to acknowledge all the authors who took the time to respond to email requests and dug up old data and scripts.

Reviewer: 2

Comments to the Author(s)

I have some minor comments in my review:

Review of paper: "Estimating the reproducibility of social learning research published between 1955 and 2018"

Summary

As the title states, this paper assessed and estimated the reproducibility of a set of papers published in social learning from 1955 to 2018.

This is an excellent paper. I enjoyed reading it immensely. The methodology is sound, and clearly a large amount of effort has been put in all aspects, from the methodology itself being very time consuming and writing and responding to many emails, to the sophisticated, but not too complex statistical model, through to the discussion that brings home many key points about the current issues in reproducibility.

I also greatly appreciated that a github repository (<https://github.com/rianaminocher/reproducibility-analysis>) was provided with the code, and in particular that time was taken to write a README to explain how to run the analysis. My only thoughts further to this are that a system like targets (<https://github.com/ropensci/targets>), and renv (<https://github.com/rstudio/renv>), could be used in the authors' future work, to help make reproducibility even more stable, as well as help improve installation. I don't want the authors to turn around and use targets and renv with this paper, but I just thought they might be interested in knowing about it.

I have only very minor comments.

Comments on the paper

Section 1, line 15

It might be useful to very briefly distinguish between the terms reproduction and replication earlier in the paper - since "replicability" is quoted in the second paragraph, perhaps that could be a good place to briefly define the two terms? I do note that replicability is defined in the third paragraph on page one, and that these terms have been defined many times in other papers, so I will understand if the authors would prefer to not do this.

Section 1, line 51

What are the different types of data? Can you provide some examples of these?

Section 2A, line 11

What does forward and backward citation tracking mean?

Section 2B, part ii, line 34

I assume that this is sample without replacement? I also assume a seed was set, but I could not find this in the GitHub repository. I guess as long as the data that was subsetted is stored from the first time this isn't a problem, so perhaps there is no need to address this in the text, but just thought it might be worthwhile to mention.

Section 2C, line 31

"We allow both alpha and lambda to vary with the species studied..."

Is "species" here a typo?

Figure 1

I found Figure 1 quite tricky to interpret. I could not understand which text belonged to which bar - for example on the left hand side, does the top bar belong to "request sent 84%"? If so, then what text belongs to the bottom bar? Is it "Data/code received 19%"? or "Data recovered 30%"?

Another reason this figure is hard to understand is that the bars are unaligned to an axis, so it is hard to compare and understand them (see <http://visiphilia.org/2016/08/03/CM-hierarchy> for more details on Cleveland-McGill's graphical hierarchy).

Since I believe the main goal of this visualisation is to communicate the flow of the number of surveyed, and where those numbers decreased from 100%, I think a sankey or alluvial chart might be useful? I would recommend potentially rebuilding it with ggalluvial (<https://cran.r-project.org/web/packages/ggalluvial/vignettes/ggalluvial.html> - an example - <https://mdneuzerling.com/post/my-data-science-job-hunt/>).

If an alluvial chart is not desirable, it would at least be helpful in the figure text to describe which way (top to bottom, bottom to top) to read the figure.

Figure 2A

Figure 2A should contain a legend to indicate what the bubble size represents.

===PREPARING YOUR MANUSCRIPT===

If you have been asked to revise the written English in your submission as a condition of publication, you must do so, and you are expected to provide evidence that you have received language editing support. The journal would prefer that you use a professional language editing service and provide a certificate of editing, but a signed letter from a colleague who is a native

speaker of English is acceptable. Note the journal has arranged a number of discounts for authors using professional language editing services (<https://royalsociety.org/journals/authors/benefits/language-editing/>).

===PREPARING YOUR REVISION IN SCHOLARONE===

-- If you have uploaded ESM files, please ensure you follow the guidance at <https://royalsociety.org/journals/authors/author-guidelines/#supplementary-material> to include a suitable title and informative caption. An example of appropriate titling and captioning

may be found at https://figshare.com/articles/Table_S2_from_Is_there_a_trade-off_between_peak_performance_and_performance_breadth_across_temperatures_for_aerobic_sc_ope_in_teleost_fishes_/3843624.

Author's Response to Decision Letter for (RSOS-210450.R0)

See Appendix A.

Decision letter (RSOS-210450.R1)

Dear Ms Minocher,

I am pleased to inform you that your manuscript entitled "Estimating the reproducibility of social learning research published between 1955 and 2018" is now accepted for publication in Royal Society Open Science.

Kind regards,
Royal Society Open Science Editorial Office

on behalf of Prof Kevin Padian (Subject Editor)
openscience@royalsociety.org

Appendix A

Response letter RSOS-210450: Estimating the reproducibility of social learning research published between 1955 and 2018

Associate Editor Comments to Author: (on behalf of Prof Kevin Padian (Subject Editor))

Thank you for your patience while the journal sought peer reviewers - regrettably, the last few months have been exceptionally difficult for editors seeking reviewers, so we are very grateful not only to the authors for your patience but the referees for your support at a tricky time.

The general view of the work is positive, and publication following revision is recommended.

One note for the authors from one of the reviewers is that there may be a minor bug in your code (see below), though the reviewer noted it may be an issue with their software, rather than the code per se (in any case, something to take a quick look at):

After following installation instructions, I could not replicate the paper and got this error:

```
...  
Error in Module(module, mustStart = TRUE) :  
function 'Rcpp_precious_remove' not provided by package 'Rcpp'  
...
```

However after restarting R this error went away, and then I got this error when running the fitting code:

```
...  
fit <- stan(file = "model.stan",  
  data = data,  
  chains = 4,  
  cores = 4,  
  iter = 1e4)  
...  
...  
...  
Chain 4: Iteration: 1 / 10000 [ 0%] (Warmup)  
Chain 4: Iteration: 4000 / 10000 [ 40%] (Warmup)  
Error in unserialize(socklist[[n]]) : error reading from connection  
...
```

Dear Professor Padian,

We appreciate the opportunity to submit a revised version of our manuscript. We would like to thank our reviewers very much for their helpful and supportive feedback, and for taking the time to review our work during this difficult period. We have addressed comments point-by-point below.

Sincerely,

Riana Minocher
PhD Student
Max-Planck Institute for Evolutionary Anthropology

Thank you for highlighting that there were issues executing the code. As a check, we have cloned the github repository on a different machine and verified that the project is reproducible, so this should not be a general issue. Regarding the first error, R needs to be restarted after installing rstan, please see the instruction page (<https://github.com/stan-dev/rstan/wiki/RStan-Getting-Started>). Unfortunately, the error “unserialize(socklist[[n]]): error reading from connection” is not familiar. From briefly googling the error, it seems this could be due to memory issues. However, executing our analysis shouldn't be a problem for a regular laptop. To troubleshoot further, can the same machine reproduce the example on the Stan installation page (<https://github.com/stan-dev/rstan/wiki/RStan-Getting-Started>)? We would recommend trying a different machine, if not.

Reviewer comments to Author:

Reviewer: 1

Comments to the Author(s)

****Disclaimer:** The authors contacted me for data for some of my past social learning research, so my responses and studies would be included in their dataset / analysis. However this would apply to everyone in the field of social learning, and I had no input into the study in any other respect, so I don't consider this a conflict of interest. **

I really like this study of the reproducibility of findings in social learning research. It contributes both to the ongoing efforts to improve the scientific rigour of the behavioural and evolutionary sciences, and to the study of social learning in humans and non-human species. The methods are robust and well-documented, the results are adequately analysed and reported, and the conclusions are constructive and coherent. It's valuable and novel to know that the biggest hurdle to reproducibility is lack of availability of data, and encouraging that if data are present then a large majority of results are reproducible (and also that data availability is improving all the time). I recommend publication, with some minor suggestions / requests for clarification (in order of appearance in the ms, not importance):

Page 2, line 45 “relevant to the field of evolutionary anthropology”: this kind of goes against the subsequent sentence which highlights the multidisciplinary relevance of social learning to other disciplines like behavioural ecology, evolutionary psychology etc. Suggest deleting or moving evolutionary anthropology to the multidisciplinary list of fields.

Thank you for pointing this out. We have deleted this portion of the sentence.

Page 3, line 11: this is in the SI but I would add here for clarity that synonymous or similar terms like “cultural transmission” were also used as search terms

Thank you for suggesting this. Added.

page 6, line 11: an aside really - I understand why the papers that were selected for analysis are unidentifiable. However, I wonder if the authors of the present paper have contacted the authors of the 4 findings that failed to match the original claim? The original authors may be willing to collaborate on a re-analysis, or correct or retract the original paper. This strikes me as an important correction to the literature, if the current authors are aware that there are (potentially) false claims in the literature that are presumably still being cited, but haven't disclosed what they are.

We have not thus far contacted authors of the 4 findings that we failed to reproduce. Corrections to the literature are important. However, when we failed to reproduce a finding, we were still able to i) comprehend and use the relevant data, and ii) attempt a reproduction of the analysis. By our definition, these results are “more” reproducible than others selected. Consider that 70% of the sample may continue to be cited, but the original data simply does not exist. Relatedly, we did not feel justified in contacting researchers who provided data we could use and comprehend, but not the researchers who provided data (or made it available online, in some cases) that doesn't actually correspond to the analysis they reported. A relevant aspect of our work overall is that we could of course collaborate further on re-analyses with each author to increase overall reproducibility (note our “reasonable researcher criterion”). Still, many authors admitted to us during correspondence that, even upon sharing data with us, their analysis is likely no longer reproducible. It seemed to get in touch, to repeat that we found the same, would not be so productive. We suggest that these processes of correction should be baked into the way we continue to produce research, by encouraging data sharing on publication, code review, the use of version control, development of data-sharing infrastructure and data management skills in individual researchers. Finally, as you note, our primary concern was protecting the identity of authors who participated in our study, especially those who shared data with us. Still, we did consider whether these particular analyses warranted retraction, with the strength of the claims of the original authors in mind. A point to consider further is that if a single result of a paper fails to reproduce, and the paper is largely cited for its contribution due to other results, this may not necessarily warrant a retraction of the original article. An even further point is that we did not critique the methods of papers we reproduced, but it may equally be that we do not necessarily believe results we were *able* to reproduce, because, for example, we felt the method or design was inappropriate. All of these remain open questions. Thank you for raising this important discussion point, and we do hope that there will be future avenues in which to discuss the most ethical and efficient ways to “correct” the scientific literature.

Page 6, line 36: regarding the differences between human, non-human, observational and experimental categories: I wonder how robust these are to author differences? E.g. if author X runs human experiments, authored several papers in this sample and included data in their papers / responded to requests, while author Y runs non-human observational studies, authored several of these in the sample, and did not include data / did not respond, then these category differences could be more to do with those individual author's practices than anything inherent about the different methods and study species. This may be particularly the case for a relatively small field with some authors publishing multiple studies within it. I

know there are paper-specific intercepts in the model, but I don't think there are author-specific intercepts. Can a bit more information be provided about how many papers were in each of these categories, and if the same author appeared across more than one paper, to get some insight into this?

Thank you for raising an insightful discussion point. The main text and Supplementary Information now contains the number of papers in each category ("human"/"non-human" and "observational"/"experimental"). The same author did appear across more than one paper, and we attempted to express the extent of this "authorship skew" in the main text, describing that a single author appeared anywhere between 1 and 49 times in our sample. We are also interested in how individual authors influence the adoption and spread of reproducible practices within our field. This is the goal of a follow-up project, now underway, in which we investigate the diffusion of "reproducibility norms" - primarily measured by data-sharing - over time within our sample, in a network-based framework. We did not include author-specific intercepts in the current model, rather assuming that the "authorship skew" reflects a true skew in the literature (some authors publish many papers, while many publish few). Thus, our reported estimates reflect this underlying skew. We believe the follow-up investigation of how reproducible practices are structured across authorship networks will help answer the question about whether these differences are inherent to the data types or study species, or rather related to different practices in various subfields, as we speculate in our discussion about the role of the "replication crisis" in psychology.

Page 7, figure 1 caption: Why were 89% CIs used rather than 95%? ... Just kidding, I know why. But seriously, it is only mentioned in the figure caption that 89% was used, while there are several CIs in the text. If all of the CIs in the text were also 89% this needs to be mentioned somewhere.

We have added a sentence to the methods section, which explains that all CIs reported correspond to the narrowest interval of the posterior that corresponds to the specified probability mass (89%). We use 89% compatibility intervals, rather than a conventional and arbitrary 95%, to avoid conflation with significance tests, which these are not.

SI: "If a study included both observational and experimental designs, we recorded it as "observational", and if a study included both human and non-human study species, we recorded it as "human"." I don't really see the logic of this, it needs some justification. If a study compares social learning in human children and chimpanzees on the same task (e.g. Dean et al. 2012), how come this is coded as 'human' and not 'non-human'? It strikes me that the chimpanzee part would present more methodological challenges than the human part, and be the bigger limiter on data analysis and recording. Perhaps a third 'human and non-human' category is needed here.

This is a valid concern, and thank you for pointing it out. We coded these studies as "human" as we felt human data may be harder to share than non-human, given ethical constraints (we touch on this briefly in the methods section). But we agree with your suggestion that chimpanzee data may in fact present more methodological challenges. We have returned to the data and computed the numbers in each subcategory. A third category

("nonhuman and human experimental) would be difficult to analyse in the model as few studies (n = 12) fit this category. We have broken down the numbers now by category in the text, but we run our analysis with our original assumption, that human data presents the limiting constraint. We have also added a robustness check to the Supplementary Information, which re-runs our analysis for the "human and non-human" studies, previously coded "human", flipped and coded as "non-human". The category-specific predicted half-lives do not change substantially, corroborating our reported findings.

Acknowledgements: it might be nice to acknowledge all the authors who took the time to respond to email requests and dug up old data and scripts.

This is a truly valuable suggestion, we apologize for overlooking this, and we now acknowledge all the authors of the work we sampled.

Reviewer: 2

Comments to the Author(s)

I have some minor comments in my review:

Review of paper: "Estimating the reproducibility of social learning research published between 1955 and 2018"

Summary

As the title states, this paper assessed and estimated the reproducibility of a set of papers published in social learning from 1955 to 2018.

This is an excellent paper. I enjoyed reading it immensely. The methodology is sound, and clearly a large amount of effort has been put in all aspects, from the methodology itself being very time consuming and writing and responding to many emails, to the sophisticated, but not too complex statistical model, through to the discussion that brings home many key points about the current issues in reproducibility.

I also greatly appreciated that a github repository (<https://github.com/rianaminocher/reproducibility-analysis>) was provided with the code, and in particular that time was taken to write a README to explain how to run the analysis. My only thoughts further to this are that a system like targets (<https://github.com/ropensci/targets>), and renv (<https://github.com/rstudio/renv>), could be used in the authors' future work, to help make reproducibility even more stable, as well as help improve installation. I don't want the authors to turn around and use targets and renv with this paper, but I just thought they might be interested in knowing about it.

Thank you very much for the suggestions.

I have only very minor comments.

Comments on the paper

Section 1, line 15

It might be useful to very briefly distinguish between the terms reproduction and replication earlier in the paper - since “replicability” is quoted in the second paragraph, perhaps that could be a good place to briefly define the two terms? I do note that replicability is defined in the third paragraph on page one, and that these terms have been defined many times in other papers, so I will understand if the authors would prefer to not do this.

Thank you for pointing this out. We do actually attempt to briefly define both terms in text, but we note that we do so somewhat ambiguously (i.e. we define replicability in the second paragraph, and reproducibility in the fourth paragraph). We now more clearly define replicability in the second paragraph, and reproducibility in the fourth paragraph using the phrase “may be defined as”. In both places, we cite Goodman et al. (2016).

Goodman SN, Fanelli D, Ioannidis JP. What does research reproducibility mean?. Science Translational Medicine. 2016.

Section 1, line 51

What are the different types of data? Can you provide some examples of these?

Yes. We have added further details.

Section 2A, line 11

What does forward and backward citation tracking mean?

Searching citation lists and “cited by” lists, of articles already sampled. We now provide a brief definition of this process in the Supplementary Information.

Section 2B, part ii, line 34

I assume that this is sample without replacement? I also assume a seed was set, but I could not find this in the GitHub repository. I guess as long as the data that was subsetted is stored from the first time this isn't a problem, so perhaps there is no need to address this in the text, but just thought it might be worthwhile to mention.

Thank you, yes. This is not in the Github repository. We don't believe this should be an issue. The script simply sampled, without replacement, from the full list of study keys with data available, and stored the list of selected study keys. We then wrote the list of sampled keys to a .txt file with the R command “write.table()”. We also used this list of study keys to simply make a “subsample” folder of data immediately, using some bash script. Because we used a single study key throughout the project to reference all materials for a particular publication, this procedure was fairly smooth. The stages are documented in our local version history, but of course not in the public repository, given the identifying information in the study keys. We hope this explanation helps, but agree it is a bit too detailed for the main text. We have added “sampled without replacement” to the main text, to be clearer.

Section 2C, line 31

"We allow both alpha and lambda to vary with the species studied..."

Is "species" here a typo?

"Species" should not appear to be a typo. What we mean is that alpha and lambda (the rate of decay and baseline probability parameters) are estimated for each category (human experimental, non-human experimental, human observational, non-human observational), therefore to an extent do "vary with species". However, thank you for pointing out that our language is not very clear, so we have now modified the explanation in text.

Figure 1

I found Figure 1 quite tricky to interpret. I could not understand which text belonged to which bar - for example on the left hand side, does the top bar belong to "request sent 84%"? If so, then what text belongs to the bottom bar? Is it "Data/code received 19%"? or "Data recovered 30%"?

Another reason this figure is hard to understand is that the bars are unaligned to an axis, so it is hard to compare and understand them (see <http://visiphilia.org/2016/08/03/CM-hierarchy> for more details on Cleveland-McGill's graphical hierarchy).

Since I believe the main goal of this visualisation is to communicate the flow of the number of surveyed, and where those numbers decreased from 100%, I think a sankey or alluvial chart might be useful? I would recommend potentially rebuilding it with ggalluvial (<https://cran.r-project.org/web/packages/ggalluvial/vignettes/ggalluvial.html> - an example - <https://mdneuzerling.com/post/my-data-science-job-hunt/>).

If an alluvial chart is not desirable, it would at least be helpful in the figure text to describe which way (top to bottom, bottom to top) to read the figure.

We thank you for your references to the Cleveland-McGill's graphical hierarchy and ggalluvial, and for pointing out that our figure is tricky to interpret. We have made modifications to the original figure, and expanded the figure caption, based on these points. First, we have made sure that each bar corresponds to a text box. We have moved the numerical proportions onto the actual bar, to make clearer which bars they correspond to. We have detailed how to interpret the figure in the caption, i.e. the left panel should be read first, top to bottom, and then the right panel, again top to bottom. We have also added some "guidelines" to the figure which we believe will help readers navigate the figure better. We simply preferred a diagrammatic representation of the "Sankey" for aesthetic reasons, which motivated our original figure. We agree that the original figure was somewhat confusing, and based on this feedback, we have now modified the figure so it now conveys all the necessary information. Thank you!

Figure 2A

Figure 2A should contain a legend to indicate what the bubble size represents.

We have added a legend to Figure 2A.